# Inflammation-Involved Proteins in Blood Serum of Cataract Patients—A Preliminary Study

**DOI:** 10.3390/biomedicines11102607

**Published:** 2023-09-22

**Authors:** Paweł Sutkowy, Hanna Lesiewska, Alina Woźniak, Grażyna Malukiewicz

**Affiliations:** 1Department of Medical Biology and Biochemistry, Faculty of Medicine, Ludwik Rydygier Collegium Medicum in Bydgoszcz, Nicolaus Copernicus University in Toruń, 85-092 Bydgoszcz, Poland; 2Department of Ophthalmology, Faculty of Medicine, Ludwik Rydygier Collegium Medicum in Bydgoszcz, Nicolaus Copernicus University in Toruń, 85-094 Bydgoszcz, Poland; hanna.lesiewska@cm.umk.pl (H.L.); g.malukiewicz@cm.umk.pl (G.M.)

**Keywords:** cataract, lysosomal enzymes, serpin, adipokines

## Abstract

Approximately 50% of all global blindness is caused by cataract in adults aged ≥50 years. The mechanisms of the disease are most arguably related to a redox imbalance and inflammation; therefore, the aim of the study was to evaluate the processes associated with inflammation in cataract patients. Twenty-four patients aged 22–60 years (62.5% females) participated in the study, with 33 controls aged 28–60 years (66.7% females). Venous blood serum of the subjects was examined for alpha 1-antitrypsin, as well as selected lysosomal enzymes and adipokines. The activities of lysosomal enzymes, as well as the activity of alpha 1-antitrypsin and the concentrations of c-reactive protein and leptin, were similar in the patients versus the controls. The concentrations of interleukin 6 and resistin were lower, in turn, whereas omentin-1 and adiponectin were higher. Moreover, the study revealed the existence of many linear relationships between the parameters, including multiple linear regression, especially gender-wise. No systemic inflammation was probably noted in the cataract patients tested; nevertheless, the deregulation of adiponectin, omentin-1 and resistin secretion was observed.

## 1. Introduction

Among the global 33.6 million adults aged 50 years and older who were blind in 2020, cataract was the leading cause of blindness (15.2 million cases). In terms of the relative contribution to the age-standardized prevalence of total blindness in adults over the age of 50 years, cataract accounted for 45.5% of blindness worldwide [1].

Cataract tends to be a multifactorial disease. Contributing risk factors for cataract development include medical conditions and complications, but also genetic, metabolic and traumatic, as well as toxic and radiation, factors. There are also cases of cataracts even at a relatively young age, in which the cause(s) of the disease cannot be clearly identified [2]. The most common cataract risk factor, however, is age. The prevalence of cataract increases with age and is higher in women than in men. In China, for example, age-related cataract develops in 50% of individuals under the age of 80 years [3]. In a Turkish study, senile cataract was found in approximately 80% of category B driver’s license holders aged over 71 years, versus 55% in the 61–70 age group [4].

There are three primary types of cataract, relative to the lesion location. Sclerosis and yellowing of the central part of the lens is typical for the most common type of cataract, i.e., nuclear cataract. Cortical cataract is characterized by grey and white, wedge-shaped opacities widening toward the lens periphery. Subcapsular cataract involves opacities located directly under the lens capsule. Patients with advanced-stage cataract complain of blurred vision, the loss of contrast, halos and difficulty with glare, which impede daily activities significantly.

It is not entirely clear what biological mechanisms are involved in the development of cataracts of unknown etiology, most probably dependent on age alone (age-related cataract) [5]. The most likely mechanism entails an imbalance between the production of reactive oxygen species (ROS) and reactive nitrogen species, as well as the elimination of these species by antioxidants (oxidative stress). Chronic inflammation may be another factor promoting cataract development [6] (Figure 1). Several studies have already implied a link between systemic inflammation and the development of age-related cataract [7,8,9]. In our previous study, enhanced lipid and protein oxidation, as well as the depletion of antioxidant defense, in cataract were observed in patients under 60 years of age [10]. In the present study, we undertook to determine the serum activities of selected lysosomal enzymes and serpin (alpha 1-antitripsin, AAT), along with the serum levels of adipokines, an inflammatory marker (c-reactive protein, CRP) and interleukin 6 (IL-6), in a similar group of relatively young patients who suffered from cataract of an unknown etiology and without any coexisting diseases, in order to verify our hypothesis stating that systemic inflammation is a potential driver of cataract development.

## 2. Materials and Methods

### 2.1. Study Subjects

The study group consisted of 24 patients (15 females and 9 males) aged 22–60 years, who were qualified for cataract surgery. A thorough eye examination with pupil dilation revealed 7 nuclear, 12 cortical and 5 subcapsular cataract cases. The visual acuity, as assessed using the Snellen chart and presented in decimal values, was equal to or worse than 0.6 in at least one eye. Bilateral cataracts were found in 17 patients, and in 7 cases, the second eye was pseudophakic. The reference group consisted of 33 individuals without cataract, matched for age (28–60 years) and gender (Table 1).

All study participants signed a voluntarily consent form to participate in the research. The study received relevant approval from the ethics committee at the Medical College of the Nicolaus Copernicus University in Torun (Bydgoszcz, Poland) (no. KB 866/18) and was conducted in accordance with the ethical principles of the Declaration of Helsinki. Patients with a history of ocular surgery or trauma, glaucoma, myopia (over 5 dioptres), age-related macular degeneration, optic neuritis and chronic inflammatory ocular conditions (e.g., chronic uveitis) were excluded from the study. The exclusion criteria also included hypertension, diabetes, rheumatoid arthritis, lipid profile disorders, autoimmune diseases, neurologic disorders, neoplasms and the chronic use of local or systemic corticosteroids and cigarette smoking. There were no diagnosed alcoholics in either group. Physical activity-taking patients, in turn, were not excluded from the sample. The general characteristics of the studied and control groups collectively are shown in part A of Table 1, whereas Table 1B divides these groups by gender.

The test material was blood serum. Fasting whole blood was drawn from basilic veins (elbow flexion) into separation gel procoagulant vacuum tubes (5 mL). Following room temperature coagulation of the blood (15–30 min), the samples were centrifuged for 10 min at 1200× *g* (4 °C). The top layer (serum), after pipetting, served as the final test material. UV/Vis spectroscopy-based assays were performed.

### 2.2. Methods

The following lysosomal enzymes were studied: acid phosphatase (AcP: EC 3.1.3.2), cathepsin D (CTS D: EC 3.4.23.5), and arylsulphatase (ASA: EC 3.1.6.1). AcP, CTS D, ASA, and AAT were determined using reagents purchased from Merck (Merck KGaA, Darmstadt, Germany).

The activity of AcP was determined using Krawczyński’s modification of Bessey’s method [11]. Disodium p-nitrophenylphosphate, dissolved in a 0.5 mol/L citrate–tartrate–formaldehyde buffer (pH = 4.9), was used as the reaction substrate. The serum was added to the substrate solution and incubated for 30 min at 37 °C (human body temperature). As a result of the enzyme activity, 4-nitrophenol (4-NP) was produced; therefore, the activity is expressed as 4-NP produced per milligram protein over a minute (nmol 4-NP/mg protein/min). The concentration of 4-NP was calculated based on the absorbance results of the following samples: control (H_2_O as a substrate solution), blank (H_2_O instead of serum) and standard (4-NP dissolved in 0.2 mol/L NaOH). The wavelength was 405 nm.

CTS D activity was assayed via the color hydrolysis of 2% denatured bovine hemoglobin at 37 °C, using a staining solution (phenolic reagent). The color intensity was directly proportional to the activity of the enzyme in the blood serum that was added to the reaction mixture. CTS D activity resulted in the production of tyrosine (TYR). Absorbance was read at 660 nm, and the final results were read from the standard curve of TYR concentrations for the same wavelength (nmol TYR/mg/min) [12].

The measurement of ASA activity was performed using Roy’s method with Błeszyński’s modification [13]. The substrate for the enzyme was 4-nitrocatechol (4-NC), in the form of a sulfate salt (0.01 mol/L) in acetate buffer (0.5 mol/L) (pH = 5.6). Higher quantities of pure 4-NC led to higher ASA activity (nmol 4-NC/mg protein/min). Absorbance readings was performed at λ = 510 nm.

AAT is a protein-inhibiting serine protease, e.g., trypsin (TR); therefore, the measuring principle consisted of decreasing the TR activity after incubation with the subject’s serum. The reaction mixture was composed of benzoyl-DL-arginine-p-nitroanilide (substrate for TR), TR solution, and 0.1 mol/L Tris buffer with HCl and 0.02 mol/L CaCl_2_ (pH = 8.2). Absorbance readings were taken at wavelength of 410 nm, along with control (no serum sample) and blank (no TR solution) samples. AAT activity in the sample was expressed as the concentration of TR retained by AAT (mg TR/mL serum) [14].

Serum concentrations of CRP, IL-6, and adipokines (resistin, omentin-1, adiponectin, leptin) were measured using commercial sandwich ELISA kits. CRP was measured using an Immundiagnostik kit (CRP ELISA, Immundiagnostik AG, Stubenwald-Allee 8a, 64625 Bensheim, Germany) and a Diaclone IL-6 kit (Human IL-6 ELISA Kit, Diaclone SAS, 6 Rue Docteur Girod, 25020 Besançon Cedex, France), whereas the adipokines were assayed using BioVendor kits (HUMAN RESISTIN ELISA, HUMAN OMENTIN-1 ELISA, HUMAN LEPTIN ELISA, HUMAN ADIPONECTIN ELISA, BioVendor Group, Karásek 1767/1, 621 00 Brno, Czech Republic). The methods relied on the use of capture antibodies coated to the 96 wells of a microplate, which proved highly specific for the analyte, detection antibodies conjugated to horseradish peroxidase (HRP), peroxidase substrate (tetramethylbenzidine, TMB), standards, quality controls, and stop solutions. A yellow reaction product indicated the presence of the analyte. The intensity of the color was directly proportional to the concentration of analyte in the sample and read from the standard curve.

The CRP concentration was determined after quenching the reaction with an acidic stop reagent based on absorbance readings at a 450 nm wavelength, and expressed as milligrams CRP per liter of serum (mg/L).

The concentrations of IL-6, resistin and omentin-1 were determined using biotinylated detection antibodies, which were combined with an HRP–streptavidin conjugate. After stopping the reaction (acidic stop solution for IL-6 and omentin-1, hydrogen peroxide for resistin), the optical density (OD) was measured at two wavelengths: 450 nm + 630 nm for IL-6 and omentin-1, and 450 nm + 620 nm for resistin. The second measurement at a higher wavelength served as the reference; thus, the difference between both measurements represented the final OD result. Ultimately, the concentrations were expressed as picograms of IL-6 per milliliter of serum (pg/mL) and nanograms of resistin/omentin-1 per milliliter of serum (ng/mL).

Adiponectin and leptin concentrations were also measured at two wavelengths: 450 nm + 630 nm. The concentrations are presented as micrograms of adiponectin or nanograms of leptin per milliliter of serum (µg/mL and ng/mL, respectively).

### 2.3. Statistical Analysis

The results of the study were statistically analyzed using parametric tests (independent *t*-test for Table 2A, and ANOVA with Tukey’s post hoc test for unequal N values with respect to Table 2B), with a prior assessment of the assumptions behind those tests (Kolmogorov–Smirnov test of normality and Levene’s test of variance homogeneity). Additionally, simple and multiple linear regressions were performed between the results and the general characteristics of the subjects, followed by power analysis with a minimum sample size analysis. Statistical significance was assumed at a p-level below 0.05. The entire statistical analysis was carried out using Statistica (version 13.3) (TIBCO Software Inc., 2300 East 14th St., Tulsa, OK, USA). The results are presented in the paper as arithmetic means and standard deviations.

## 3. Results

The results of the study, along with statistically significant differences and correlations, are shown in Table 2 and Table 3, respectively. Both the simple and multiple linear regressions suggested a significant effect of gender on the parameters tests, although the numerical ratio of females to males was similar (62.5%:37.5% in the patients, and 66.7%:33.3% in the controls); therefore, we decided to divide the results into gender-specific subgroups, including Table 3 showing the correlations (Table 2B and Table 3B). Multiple stepwise linear regression (predictors: age, gender, BMI) showed the following linear relationships: gender vs. AAT concentration: r = −0.28 (*p* = 0.045), as well as gender vs. the adiponectin concentration: r = −0.46 (*p* = 0.0005).

No statistically significant differences between the entire groups were found for lysosomal enzymes, AAT, CRP and leptin. The IL-6 and resistin concentrations, in turn, showed higher levels in all controls, i.e., by 30% (*p* < 0.05) and 64.9% (*p* < 0.001), respectively, compared to the cataract patients. The omentin-1 and adiponectin concentrations were lower in the controls (35.2% at *p* < 0.001 and 22.3% at *p* < 0.05, respectively).

After dividing the groups by gender, the AcP and omentin-1 concentrations were higher and resistin concentrations were lower in female cataract patients, versus healthy female controls. Likewise, the IL-6 and resistin concentrations were lower, while the concentrations of omentin-1 and adiponectin were higher, in females with cataract, relative to the levels measured in male controls. The ASA activity and IL-6 concentrations were lower and leptin concentrations were higher in control females, compared to both the cataract-group and control males. The activity of AAT and the concentration of resistin, in turn, were higher in the control females only, relative to those in the males with cataract, whereas the adiponectin concentrations were higher only in the control females, compared to the control males. Statistically significant differences were also found between cataract and healthy males for the measurements of IL-6, resistin and adiponectin concentrations—the IL-6 and resistin concentrations were lower, and the adiponectin levels were higher in the group of cataract patients.

The study furthermore revealed, through a simple linear regression, a number of statistically significant linear correlations between the study results and the general characteristics of the subjects. Due to the large number, as mentioned, the correlations with gender merit special attention, although age also correlated significantly with many of the parameters (Table 3). The most noteworthy, in our opinion, correlations are presented in the form of scatterplots (Figure 2, Figure 3, Figure 4 and Figure 5).

Given the limited number of participants in the study, a minimum sample analysis (N) was estimated. Indicatively (based on Table 2A), the N required for a single study group at 80% power of the test (independent *t*-test), alpha = 0.05 (statistical significance threshold), and sigma (population SD) as the mean/median of the SDs obtained in the study, along with/per actual powers, were as follows: AcP = 78/38%, CTS D = 2553/6%, ASA = 75/39%, AAT = 172/20%, CRP = 3115/6%, IL-6 = 24/86.2%, resistin = 7/100%, omentin-1 = 10/100%, adiponectin = 31/77%, leptin = 149/2%.

## 4. Discussion

The present study demonstrated lower concentrations of IL-6 in cataract patients (the entire group), versus the control group of healthy subjects. Taking the division by gender into account, lower concentrations of IL-6 were noted only in the males with cataract, compared to the control-group males, with the control concentration of IL-6 in women lower than in men. A positive linear relationship between the IL-6 concentration and gender was also found between these parameters in the control group (r = 0.47, *p* = 0.041; Figure 2). It is difficult to interpret the results unequivocally. The literature demonstrates that, in healthy subjects, the blood level of IL-6 ranges between 1 and 5 pg/mL [15], whereas during inflammation, it increases up to several thousand times [15]. In the present authors’ own study, the serum level of IL-6 in the cataract patients totaled 5 pg/mL, whereas in the healthy controls, it showed a statistically significantly higher concentration, with a difference of only 30%, however.

IL-6 is a cytokine with broad-spectrum biological activity; it serves both pro-inflammatory and anti-inflammatory functions [16]. This biphasic/opposite IL-6 activity can be explained by the hormesis theory, according to which the effects of cytokine activity are dependent on its concentration and persistence time [17]. The levels of IL-6 may have been influenced by adiponectin, for which serum levels were higher in cataract patients, compared to in controls. The most recent studies show that adiponectin can inhibit inflammatory markers, e.g., tumor necrosis factor alpha, CRP or IL-6 [18]. The lower levels of IL-6, observed in the cataract patients, may also result from reduced physical activity, which has been also suggested by other authors [19]. Since cataract is negatively correlated with visual functions [20], IL-6 released in the muscles as a result of exercise enters the blood [17]. It seems, however, that the effect of AAT on IL-6 levels can be ruled out, as no statistically significant differences in the serum activity of this protein were found in this study, while, it has already been shown that AAT, as an inhibitor of serine protease in the blood, may inhibit IL-6 [21].

Other authors have observed no differences in serum levels of IL-6 in patients with cataracts (aged over 60 years), compared to in healthy individuals [22]. An increase in the level of IL-6 was confirmed in the post-cataract surgery aqueous humor, however [23,24]. This local IL-6 concentration can increase over 4000 times, whereas the concentration in blood serum can be below detection limits at the same time [24]. Moreover, virtually no differences of IL-6 concentrations were observed in the tear fluid of the cataract patients and the healthy controls [25].

IL-6 plays a major role in the induction of CRP expression [26], which is one of the acute-phase proteins [27]. This protein is considered a marker of systemic inflammation [7]; it is produced in the liver and released into the bloodstream in response to inflammation [28]. In the present study, virtually no differences were found in serum CRP levels, neither in cataract patients nor in healthy controls. The results obtained therefore do not confirm that cataracts in relatively young persons is related to systemic inflammation. Other authors also demonstrated that the prevalence of nuclear, cortical and subcapsular cataracts in Chinese patients aged over 50 years was not significantly associated with the serum CRP concentration [29]. Multivariate analysis (multivariate-adjusted odds ratio) also failed to confirm a correlation between serum CRP levels and the presence of cataract in Malay patients aged between 40–80 years [30]. Higher levels of CRP were in turn observed in the blood plasma of males who later developed cataracts, compared to in those who remained cataract-free [7].

The present study showed no statistically significant differences regarding the activities of lysosomal enzymes (AcP, CTS D, ASA) determined in the whole groups of cataract patients and healthy controls. Taking the division by gender into account, statistically significantly higher AcP activity was demonstrated in the females with cataract, versus the female controls. Acid hydrolases are involved in the process of tissue remodeling and reorganization, possibly in a variety of pathological conditions of the lens, as well [31], similarly to other lysosomal enzymes. It is currently believed that lysosomes play a key part in maintaining cellular and tissue health [32]. They can regulate inflammation both positively and negatively [33]. Lysosomal enzymes, in small amounts, constantly leak from lysosomes into the extracellular space and then into the blood. This increased release may, in turn, be selective and may indicate tissue remodeling and/or pathological processes [34]. The present study demonstrated a positive correlation between serum activities of CTS D and ASA in cataract patients (r = 0.43, *p* = 0.039; Figure 3A) and between serum activities of AcP and ASA in healthy controls (r = 0.35, *p* = 0.046; Figure 3B), which may suggest the nonselective permeation of enzymes from lysosomes. The available literature lacks studies similar to the one presented in this paper. Hiraoka et al. [35] have demonstrated, however, that in the aqueous humor obtained from patients with senile cataract but no other ocular diseases, lysosomal phospholipase A2 activity was lower than in the aqueous humor from patients with cataract accompanied by chronic uveitis.

The authors’ present study showed higher serum concentrations of adiponectin in cataract patients (the entire group), versus healthy controls. In analyzing the levels of adiponectin in subgroups by gender, a higher concentration was found in males with cataract, compared to in control males. In parallel, the levels of adiponectin in females with cataract were higher than in males with cataract. Moreover, multiple linear regression (predictors: age, gender, BMI) showed that the level adiponectin was negatively correlated with gender (r = −0.46; *p* = 0.0005), similarly to simple linear regression (r = −0.44, *p* = 0.046 in the cataract patients, r = −0.64, *p* = 0.001 in the controls; Figure 4). These results are in line with the studies by other authors, who have also proved that the adiponectin concentration is higher in females than in males [36,37], despite the negative association between estradiol and adiponectin [36]. Adiponectin, which is the main circulating adipokine, can have both an anti-inflammatory [38,39] and pro-inflammatory properties [40]. It is mainly secreted by adipose tissue [39], but also locally in the brain and retina [41]. Higher adiponectin levels were also demonstrated, for instance, in patients with rheumatoid arthritis, compared to in healthy controls [42]. Reduced levels of adiponectin have been observed in patients with diabetes, metabolic syndrome, obesity, coronary heart disease and hypertension [43]. Adiponectin is believed to exert protective effects in eye diseases [41].

A higher level of omentin-1 and lower level of resistin were also shown in this study, in the entire group of cataract patients. In analyzing the differences in the gender-specific concentrations of these adipokines, a higher omentin-1 concentration was observed only in women with cataract women, relative to in healthy females, while resistin concentrations were lower in both females and males, in comparison with the respective control subgroups. Omentin-1 is an adipokine with anti-inflammatory activity [44], produced mainly in visceral and epicardial adipose tissue [45]. Its relevance to the issues discussed in this paper has been confirmed based on the demonstrated negative correlation between age and the omentin-1 concentration in the controls (r = −0.54, *p* = 0.008), including the males from this group only (r = −0.78, *p* = 0.023) (Figure 5). Resistin, in turn, exerts pro-inflammatory effects, modulating metabolism and inflammation during aging [46]. In humans, it is subject to expression mainly in leukocytes (monocytes, neutrophils, macrophages) [47,48]. In summary, some distortion of adipokine secretion in the cataract patients has been demonstrated. Due to the diversity of adipokine-related signaling mechanisms, it is difficult to interpret the differences observed unequivocally. The interpretation is also compromised by the fact that not all functions of the aforementioned adipokines are likely to have been recognized. It seems that the differences shown may result from compensatory changes associated with the ongoing process of disease in an organism. Other authors, for example, explain an increase in serum adipokine concentrations in Alzheimer’s patients by a systemic compensatory mechanism acting against neurodegeneration [18]. The observed group-to-group disparities may also be attributable to the varied physical activity of cataract patients and control subjects. The long-term effects of physical activity on inflammation have been shown to be evident directly in adipose tissue and are related to effects on adipokine secretion [49].

The available literature lacks publications comparing serum concentrations of adiponectin, omentin-1 and resistin in cataract patients and healthy subjects. At most, the levels of these adipokines are compared in cataract patients with other eye diseases. For instance, no differences in plasma adiponectin concentrations have been demonstrated, when comparing senile cataract patients with patients suffering from proliferative diabetic retinopathy. In the aqueous humor, however, adiponectin levels were significantly higher in patients with proliferative diabetic retinopathy [50]. The levels of proteins determined in aqueous humor do not necessarily correlate with the plasma or serum levels thereof. This is attributable to the existence of the blood–retinal barrier providing intraocular homeostasis [50]. Patients with senile cataracts, in turn, have been shown to have lower levels of resistin in the aqueous humor, relative to those in patients with retinal vein obstruction [51].

Reports have already been published, suggesting the involvement of leptin in the mechanisms underlying the development of age-related cataracts [52]. This involvement can be explained by its contribution to the development of cataracts through the induction of oxidative stress [53]. The authors’ own study presented in this paper, however, has shown no statistically significant differences in serum levels of leptin in cataract patients and healthy controls. Noteworthy is the higher concentration of leptin in healthy females, relative to in healthy men. Gender influence has been also suggested by the negative correlation between the level of leptin and gender in the whole control group (r = −0.58, *p* = 0.039). Moreover, the leptin levels were positively correlated with the CRP levels in all cataract patients (r = 0.83, *p* = 0.011) (Table 3A), including females (r = 0.95, *p* = 0.015; Table 3B). BMI in turn, was negatively correlated with the levels of leptin in males with cataract (r = −0.99, *p* = 0.019), similarly to adiponectin (r = −0.99, *p* = 0.049). The interpretation of these dependencies is not, however, entirely evident and unambiguous, which confirms the need to conduct further research. Nevertheless, studies showing that reduced blood levels of leptin lead to increased food intake [54] and higher BMIs increase the risk of cataracts have already been published [55].

One certain limitation of the present study consists of the scant number of participants, which is why the results obtained should be treated as preliminary findings, provoking further research. The limited amount of venous blood available for collection also affected the number of parameters indicated. Most certainly, a larger volume would have enabled the authors to conduct a deeper analysis of the differences observed between the patient group and healthy controls.

The findings of this study are verifiable with similar studies based on a comparable but larger group of patients (cataract without a known cause only). Further research into the potential involvement of venous blood biomarkers in the development of cataracts may provide valuable data on the molecular basis of the occurrence of this disease. If biomarkers in the peripheral blood, e.g., adipokines, prove to be reliable predictors of cataract, patients could benefit from being informed, monitored and referred for surgical treatment sooner. This could also prove invaluable as an opportunity to develop preventive procedures.

## 5. Conclusions

Based on the results, it is likely that cataracts of unknown etiology are not accompanied by systemic inflammation. Nonetheless, the deregulation of adiponectin, omentin-1 and resistin secretion appears to have been demonstrated.

## Figures and Tables

**Figure 1 biomedicines-11-02607-f001:**
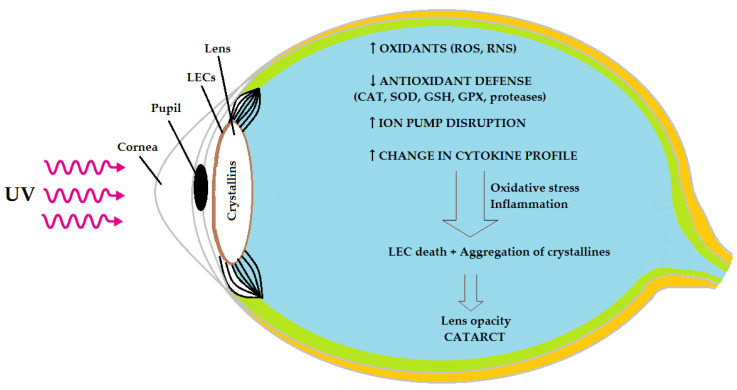
Oxidative stress and inflammation as possible causes of age-related cataract (based on [5,6,9]). **↑**: increase; ↓: decrease; LECs: lens epithelial cells; ROS: reactive oxygen species; RNS: reactive nitrogen species; CAT: catalase; SOD: superoxide dismutase; GSH: glutathione; GPX: glutathione peroxidase.

**Figure 2 biomedicines-11-02607-f002:**
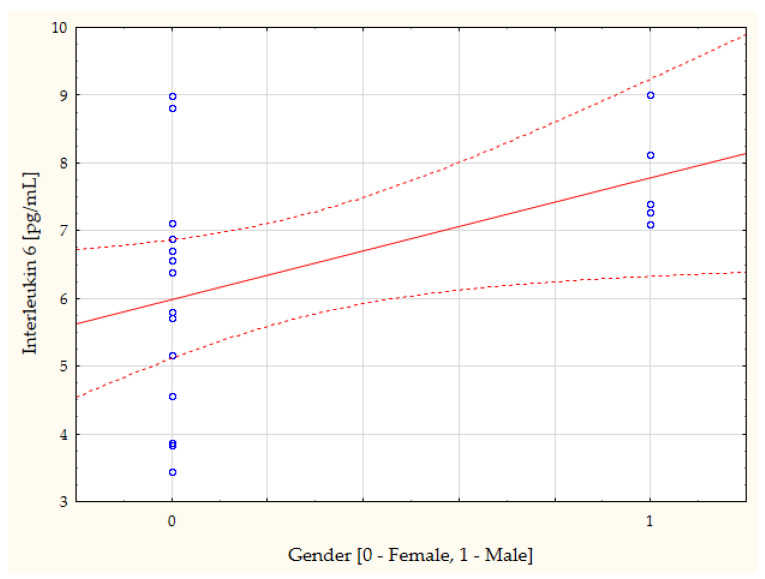
Person’s correlation coefficient between the control group concentration of interleukin 6 and gender (r = 0.47, *p* = 0.041).

**Figure 3 biomedicines-11-02607-f003:**
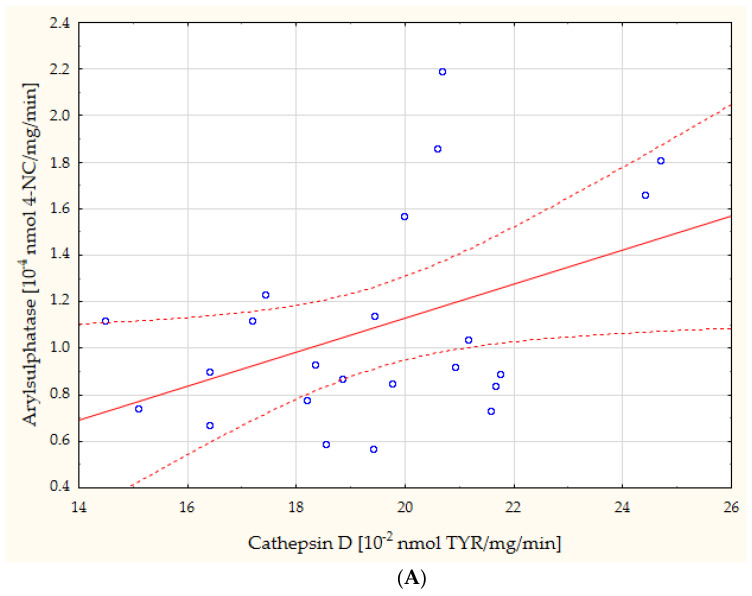
Direct linear relationships between the serum activities of CTS D and ASA in cataract patients ((**A**): r = 0.43, *p* = 0.039) and between the serum activities of AcP and ASA in healthy controls ((**B**): r = 0.35, *p* = 0.046). 4-NC: 4-nitrocatechol; TYR: tyrosine; 4-NP: 4-nitrophenol.

**Figure 4 biomedicines-11-02607-f004:**
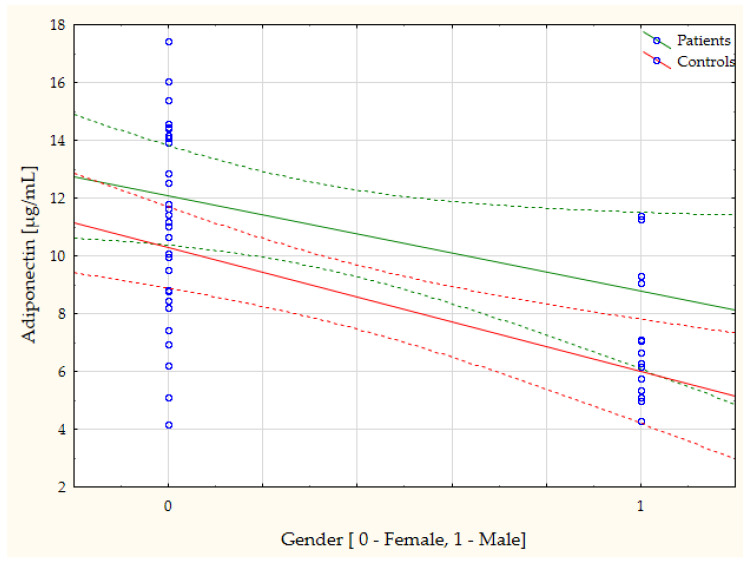
The serum adiponectin concentration is negatively correlated with gender in cataract patients (r = −0.44, *p* = 0.046) and in healthy controls (r = −0.64, *p* = 0.001).

**Figure 5 biomedicines-11-02607-f005:**
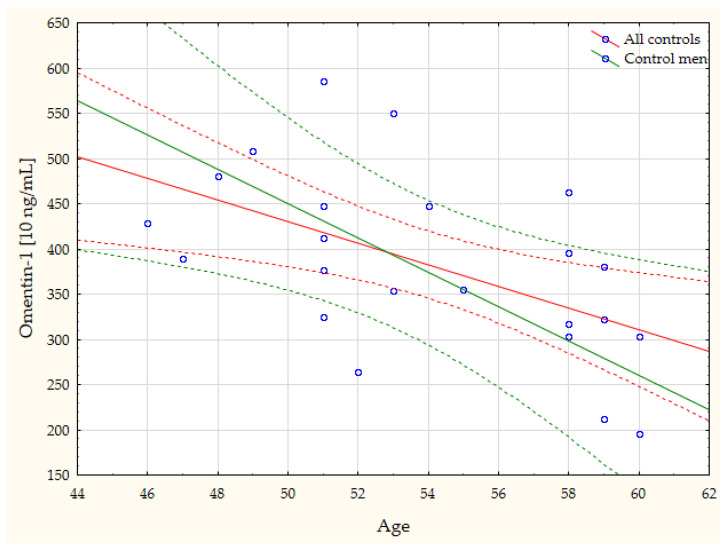
Negative linear correlations between age and omentin-1 concentrations in the controls (r = −0.54, *p* = 0.008), including the males in this group only (r = −0.78, *p* = 0.023).

**Table 1 biomedicines-11-02607-t001:** General characteristics of cataract and control groups (means ± SDs).

Part A														
					Cataract Patients					Control Group				
N					24					33				
Age (years)					52.7 ± 9.9					51.1 ± 8.2				
Sex (F/M)					15/9					22/11				
BMI					26.5 ± 2.1					25.9 ± 1.2				
**Part B**														
			**Cataract females**			**Cataract males**			**Control females**			**Control males**		
N			15			9			22			11		
Age (y/o)			53.0 ± 10.2			52.1 ± 9.9			50.6 ± 9.1			52.2 ± 6.5		
BMI			26.0 ± 2.3			27.3 ± 1.8			25.9 ± 1.3			26.0 ± 1.0		

**Table 2 biomedicines-11-02607-t002:** Serum activities of selected lysosomal enzymes and α1-antitrypsin, as well as the concentration of selected adipokines in cataract patients, in the entire groups (A) and gender-specific subgroups (B) (mean ± SD).

(A)				
Parameter	Cataract Patients		Controls	
Acid phosphatase [10^−4^ nmol 4-NP/mg/min]	11.0 ± 2.9		9.6 ± 3.3	
Cathepsin D [10^−2^ nmol TYR/mg/min]	19.4 ± 2.6		19.6 ± 2.5	
Arylsulphatase [10^−4^ nmol 4-NC/mg/min]	10.7 ± 4.4		9.2 ± 2.1	
Alpha 1-antitrypsin [10^−1^ mg TR/mL]	10.7 ± 1.9		11.2 ± 1.4	
C-reactive protein [10^−1^ mg/L]	26.8 ± 10.3		27.4 ± 6.6	
Interleukin 6 [pg/mL]	5.0 ± 1.9		6.5 ± 1.7 ^×^	
Resistin [ng/mL]	7.4 ± 2.1		12.2 ± 3.3 *	
Omentin-1 [10 ng/mL]	59.1 ± 20.2		38.3 ± 9.9 *	
Adiponectin [µg/mL]	11.2 ± 3.5		8.7 ± 3.3 ^×^	
Leptin [10^−1^ ng/mL]	56.5 ± 24.6		64.2 ± 22.6	
**(B)**				
**Parameter**	**Cataract females (1)**	**Control females (2)**	**Cataract males (3)**	**Control males (4)**
Acid phosphatase [10^−4^ nmol 4-NP/mg/min]	11.4 ± 3.2 ^a^	9.0 ± 3.6	10.2 ± 2.2	10.7 ± 2.2
Cathepsin D [10^−2^ nmol TYR/mg/min]	20.2 ± 2.8	19.5 ± 2.0	18.3 ± 1.9	19.9 ± 3.4
Arylsulphatase [10^−4^ nmol 4-NC/mg/min]	10.3 ± 4.2	8.5 ± 1.6 ^de^	11.4 ± 3.7	10.6 ± 2.4
Alpha 1-antitrypsin [10^−1^ mg TR/mL]	11.2 ± 1.8	11.5 ± 1.4 ^d^	10.0 ± 1.9	10.7 ± 1.3
C-reactive protein [10^−1^ mg/L]	28.4 ± 11.9	26.1 ± 8.1	23.4 ± 5.3	29.1 ± 4.6
Interleukin 6 [pg/mL]	5.4 ± 2.2 ^c^	6.0 ± 1.7 ^de^	4.0 ± 0.7 ^ff^	7.8 ± 0.8
Resistin [ng/mL]	7.6 ± 2.5 ^aacc^	12.6 ± 3.8 ^dd^	7.0 ± 1.3 ^ff^	11.2 ± 1.7
Omentin-1 [10 ng/mL]	62.2 ± 20.8 ^aacc^	38.2 ± 8.0	51.3 ± 18.1	38.6 ± 13.5
Adiponectin [µg/mL]	12.1 ± 3.5 ^bccc^	10.3 ± 3.1 ^ee^	8.8 ± 2.2 ^f^	6.0 ± 1.4
Leptin [10^−1^ ng/mL]	60.1 ± 24.4	80.0 ± 7.1 ^de^	47.2 ± 19.5	54.3 ± 23.6

(A) Statistically significant differences: ^×^
*p* < 0.05, * *p* < 0.001. 4-NP: 4-nitrophenol; TYR: tyrosine; 4-NC: 4-nitrocatechol; TR: trypsin. (B) Statistically significant differences: ^a^ 1 vs. 2 (^a^
*p* < 0.05, ^aa^
*p* < 0.001), ^b^ 1 vs. 3 (*p* < 0.01), ^c^ 1 vs. 4 (^c^
*p* < 0.05, ^cc^
*p* < 0.01, ^ccc^
*p* < 0.001), ^d^ 2 vs. 3 (^d^
*p* < 0.05, ^dd^
*p* < 0.001), ^e^ 2 vs. 4 (^e^
*p* < 0.05, ^ee^
*p* < 0.001), ^f^ 3 vs. 4 (^f^
*p* < 0.05, ^ff^
*p* < 0.001).

**Table 3 biomedicines-11-02607-t003:** Statistically significant Pearson product-moment correlation coefficients (r): A—in the entire groups, B—in subgroups by gender.

(A)			
Cataract Patients		Controls	
Gender vs. Adiponectin	r = −0.44, *p* = 0.046	Age vs. Omentin-1	r = −0.54, *p* = 0.008
AAT vs. Omentin-1	r = 0.43, *p* = 0.05	Gender vs. IL-6	r = 0.47, *p* = 0.041
CTS D vs. ASA	r = 0.43, *p* = 0.039	Gender vs. ASA	r = 0.48, *p* = 0.005
CRP vs. Leptin	r = 0.83, *p* = 0.011	Gender vs. Adiponectin	r = −0.64, *p* = 0.001
IL-6 vs. Adiponectin	r = 0.80, *p* = 0.001	Gender vs. Leptin	r = −0.58, *p* = 0.039
		AcP vs. ASA	r = 0.35, *p* = 0.046
		AAT vs. ASA	r = −0.42, *p* = 0.014
		ASA vs. Resistin	r = −0.43, *p* = 0.040
**(B)**			
**Females with cataract**	**Control females**	**Males with cataract**	**Control males**
CRP vs. Leptin: r = 0.95, *p* = 0.015	Adiponectin vs. CTS D: r = −0.69, *p* = 0.005	BMI vs. Leptin: r = −0.99, *p* = 0.019	Age vs. Omentin-1: r = −0.78, *p* = 0.023
IL-6 vs. Adiponectin: r = 0.89, *p* = 0.001	Resistin vs. CTS D: r = 0.51, *p* = 0.043	AAT vs. Omentin-1: r = 0.82, *p* = 0.046	Age vs. Adiponectin: r = 0.76, *p* = 0.017
	Adiponectin vs. CRP: r = −0.89, *p* = 0.043	Adiponectin vs. Leptin: r = −0.99, *p* = 0.049	
	Resistin vs. CRP: r = 0.85, *p* = 0.031		
	Resistin vs. Adiponectin: r = −0.62, *p* = 0.017		
	ASA vs. IL-6: r = −0.58, *p* = 0.028		

CRP: c-reactive protein, IL-6: interleukin 6, CTS D: cathepsin D, ASA: arylsulphatase, BMI: body mass index, AAT: alpha 1-antitrypsin.

## Data Availability

Data supporting reported results can be found at the Department of Ophthalmology, University Hospital no. 1, Bydgoszcz, Poland.

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
