# Peer review of "Inflammation-Involved Proteins in Blood Serum of Cataract Patients—A Preliminary Study"

_biomedicines, 2023, doi:10.3390/biomedicines11102607_

Round 1

Reviewer 1 Report

The paper entitled “Is cataract in patients below 60 years of age associated with changes of serum concentration /activity of proteins involved in inflammatory processes, including adipokines” is a study based on the potential role of adipokines and other factors in the development of early onset cataract formation.

The study is interesting, yet the paper has several limitations.

The paper is difficult to read and not formatted properly. The authors should be invited to use the MDPI template and follow the correct formatting for this journal.

The figures and table are useful, and informative, and help to explain the findings, however, need to be reformatted.

The title is too long and overwinded. A catchy title that does report the hypothesis is preferred.

The exclusion criteria need to include the history of ocular trauma, chronic use of local or systemic cortisone use, and a long list of all potential causative factors that can explain the early onset of cataract formation, which the authors need to include and briefly explain.

The cohort of patients is very small and not sufficient to draw general conclusive remarks. The results should be presented as preliminary. Conclusions need to be toned down considerably.

It should be mentioned how and what kinds of future studies might be able to validate the conclusions of this paper. If a potential biomarker in the blood serum turns out to be a reliable predictor of cataract progression, the authors should explain how the results of these parameters may be helpful in the care of patients and how this may modify therapy options. In a typical clinical context, a flowchart of clinical evaluations and therapies based on individual findings could be helpful and included in the Discussion section.

The paper does not read well. To improve the English and flow of the content, extensive editing is required by a native English doctor.

Reviewer 2 Report

In the current study authors made attempt to determine the mechanism(s) of cataract development in patients <60 yrs of age. Key redox enzymes such as acid phosphatase, cathepsin D, and arylsulphatase activity and inflammatory molecules including CRP , IL 6, adipokines, resistin, omentin-1, and leptin and anti-inflammatory molecule such as adiponectin were measured in serum isolated from peripheral venous blood of both age matched control and cataract patients covering both sexes. The study provided evidence showing controls. Lower serum concentrations of interleukin 6 and resistin and higher concentration of omentin-1 and adiponectin in the patients.

1.       The basic drawback of the studies lies on its design and concept. Lens is avascular and unlike retina which is highly metabolically active, lens by design has low metabolic demand and it is not quiet established about whether a linear relationship exists between serum and aqueous humor concentration of the studied enzymes, inflammatory and anti-inflammatory molecules as reported in the current study. A recent study indicated serum and aqueous humor adiponectin levels correlate well with  diabetic retinopathy development and progression (PLoS One. 2021; 16(11): e0259683). In order to validate the current findings reported in the present study authors need to provide conclusive evidence showing a linear relationship does exist in all enzymes and biomolecules studied.

2.       The next major drawback arises from the low sample size enrolled in the current study. Enrolling only 24 patients to conclusively determine any biological effect appears to be a big challenge and the authors are advised to perform a Power analysis to determine minimum number of enrollee required in order to make any firm conclusion based on the current study design.

3.       Part A in both Tables 1 and 2 could be deleted as these are not adding any additional information. Part B on both tables would be sufficient. The comparison between gender is certainly important but just for sake of comparison some data reported in the tables do not make any sense. Comparison between cataracts female vs control male and cataracts male vs control female appear unnecessary and may be deleted. In Table 3, the legends for A and B have been switched.

4.       The discussion section of the manuscript is unusually long and does not appear drafted well. This section needs to be shortened, concise and more aligned to the current hypothesis of the study discussing the relevance of the experimental results in comparison with the current knowledge in the specific subject area.

5.       I recommend adding a schematic diagram (cartoon) showing key pathways (redox and inflammatory) involved and the functional role(s) of the enzymes and biomolecules studied in relation to cataract. Limiting the conclusion of the study up to certain age group does not appear very compelling with such a smaller study participants. I recommend removing the age parameter from the study and describe the finding as more of a universal age-related cataract perspective.

Reviewer 3 Report

''Is cataract in patients below 60 years of age associated with changes of serum concentration/activity of proteins involved in inflammatory processes, including adipokines?''

The manuscript presented for review consists of 23 pages. The manuscript is divided into 5 sections.  The literature review is sufficient. Most of the references are recent. They are adequate and refer to the whole context of the study. I would recommend to modify the title (more general).

Abstract: 

- It is too long (237). According to guidelines - ''The abstract should be a total of about 200 words maximum.''.

-The structure is clear (background, methods, results and conclusions).

-Line 16: According to the definition ''presenile'' - is a period of human life between 60-73 years of age. Patients who took part in this study were 22-60 years of age...

Methods are routine and logical. Statistical tests are correct.

Conclusion is appropriate. 

-style and grammar structures.

Round 2

Reviewer 1 Report

The authors have addressed the issues in a satisfactory manner.

Moderate editing of English language is still required.

Reviewer 2 Report

The quality of the revised manuscript has been much improved. The authors have taken adequate care addressing my questions and concerns.